# Successful Practices in Performing and Maintaining Physical Activity in Adults with Type 2 Diabetes Mellitus: A Qualitative Study

**DOI:** 10.3390/ijerph192114041

**Published:** 2022-10-28

**Authors:** Mireia Vilafranca Cartagena, Antonia Arreciado Marañón, Eva Artigues-Barbera, Glòria Tort-Nasarre

**Affiliations:** 1Department of Nursing, Faculty of Health Science and Welfare, University of Vic-Central University of Catalonia (UVIC-UCC), Av. Universitària 4-6, 08242 Manresa, Spain; 2Althaia Foundation, C/Dr Joan Soler 1-3, 08243 Manresa, Spain; 3Department of Nursing, Faculty of Medicine, Autonomous University of Barcelona, Campus Bellaterra, 08193 Cerdanyola del Vallès, Spain; 4Multidisciplinary Research Group in Health and Society (GREMSAS), (2017 SGR 917), 08007 Barcelona, Spain; 5Balafia Primary Care Center, Av. de Rosa Parks. Gerència Territorial Lleida, Catalan Health Institute (ICS), 08007 Barcelona, Spain; 6Research Support Unit Lleida, Fundació Institut Universitari per a la Recerca al’Atenció Primària de Salut Jordi Gol i Gurina (IDIAPJGol), Rambla Ferran 44, 25007 Lleida, Spain; 7Department of Nursing, Faculty of Nursing and Physiotherapy, University of Lleida, C/Montserrat Roig, 25198 Lleida, Spain; 8SAP ANOIA, Gerència Territorial Catalunya Central, Catalan Health Institute (ICS), 08007 Barcelona, Spain; 9AFIN Research Group and Outreach Centre, Autonomous University of Barcelona, Campus Bellaterra, 08193 Cerdanyola del Vallès, Spain

**Keywords:** diabetes mellitus, type 2, exercise, motivations, nurses, primary healthcare, qualitative research, public health practice

## Abstract

Physical activity has proven to be greatly beneficial in patients with DM2. However, few adhere to physical activity recommendations and are motivated to engage in regular physical activity and the numerous interventions conducted to change their habits tend to be unsuccessful in the long term. The objective in this research is to study patients who adhere to physical activity in order to guide patients who have not succeeded in making this change, assisted by the successful tools in the context of community nursing. A qualitative descriptive study was conducted. The sample included 10 patients with type 2 diabetes mellitus who adhered to physical activity in Catalonia, Spain, having been selected using intentional sampling. We recorded semi-structured interviews with the participants and conducted a thematic analysis. Five categories were identified and grouped into two themes: (1) Integrate physical activity as a lifestyle (meaning of DM2 and physical activity, adaptation to change and strategies on a day-to-day basis, physical activity) and (2) Find support to change physical activity (company and personal motivational strategies). In conclusion, a good adherence to physical activity was achieved thanks to motivation maintained over time related to autonomous motivation and the psychological and physiological benefits that physical activity provides patients, external support and encouragement, and the allocation of time to adhere without making great lifestyle changes.

## 1. Introduction

Type 2 diabetes mellitus (DM2) represents a great challenge for health worldwide. According to the International Diabetes Federation, 463 million people worldwide had diabetes in 2019, 90 to 95% of whom had DM2 [1]. It has been forecast that global DM2 prevalence will increase to 7079 per 100,000 by 2030 and 7862 per 100,000 by 2040 [2]. Although DM2 is not a curable disease, modification in lifestyle can slow it down and reverse its effects [3]. The treatment for DM2 includes a correct adherence to physical activity (PA), a healthy diet and adherence to the medication prescribed. Behaviours such as a bad diet or non-performance of physical exercise represent important risk factors, and lifestyle is therefore considered the most important aspect to overcome the DM2 epidemic [1]. Daily exercise, or at least not allowing more than two days to elapse between exercise sessions, is recommended to enhance insulin action. Adults with type 2 diabetes should ideally perform both aerobic and resistance exercise training for optimal glycaemic and health outcomes. The structured lifestyle interventions should include at least 150 min/week of PA [4]. The benefits of PA include a greater sensitivity to insulin, the prevention of risk factors of cardiovascular diseases, body weight loss and an improvement in the quality of life, as well as benefits related to emotional, psychological and social spheres. Furthermore, there is a close connection between physical inactivity, the appearance of chronic complications and the development of multiple cardiovascular risk factors in individuals with DM2. Physical inactivity has been recognised as one of the predictors of death in people with DM2 [5]. Nevertheless, the perceived necessity to increase PA among diabetic patients is low [6] and they show minimal awareness of its causes, complications and treatment [7]. The effort entailed by the modification of previously established habits and a willingness to maintain them allows us to confirm that the treatment is complex, especially knowing that adults are resistant to change [8]. Previous studies have shown that long-term maintenance of weight loss and complete adherence to diet and PA recommendations is rare [9,10,11], especially in the adult population [8]. These studies show that adherence to PA is very scarce in patients with DM2 and that few patients are motivated to engage in regular PA despite the numerous interventions designed to change this, which are unsuccessful in the long term. It should be highlighted that motivation and barriers are key elements to improve adherence to the treatment of diabetes [12].

Primary care and the restructuration of the community to achieve an active lifestyle and healthy dietary habits are of utmost importance [13]. Health professionals should have both the time and resources to collaborate in educating patients individually or in groups [2]. Nurses are one of the key pillars in education and behavioural changes, and in helping to overcome the barriers people face when adopting lifestyle changes [14].

Due to the relative lack of success in community interventions and the need to clarify the interactions between some protective and risk factors related to the process of change and maintenance of PA in patients with DM2, the following questions were raised: How do diabetic patients achieve a good adherence to PA? What are the motivations for maintaining the practice of PA? Which factors facilitate or hinder the initiation and maintenance of PA? We therefore studied those patients who adhere to PA in order to provide patients who do not succeed to make this change, assisted by the successful tools in the context of community nursing.

The aim of this qualitative study was to thoroughly explore how this change in habit in the performance of PA occurs in adults with DM2 in a community environment, as well as identify how the motivation for performing PA is maintained.

## 2. Materials and Methods

A qualitative study was conducted to gain insight into participants’ individual experiences. According to the constructivist paradigm of Guba and Lincoln [15], vicarious experience, which consists of learning from observing successful behaviours, is fundamental to understanding diabetic patients with good habits in order to help non-adherers.

Initially, a deductive approach was used to develop the interview guide and an inductive analysis of the resulting data was then conducted. The current paper is part of a study on the experience of patients with diabetes during the COVID-19 pandemic [16].

The participants were patients with DM2 who practised regular PA in Catalonia (Spain); they were selected using purposeful sampling based on pragmatic and convenience criteria [17]. We recruited participants of different ages, genders and sociocultural status in order to obtain as diverse a set of narratives as possible [18]. The participants were patients with DM2 from four primary health centres in central Catalonia who were participating in our ongoing study on diabetes and physical activity. The inclusion criteria were adults aged 55 to 79 years who had been diagnosed with DM2 at least two years previously. We chose this age range because 55 is the age at which the prevalence of DM2 begins to increase rapidly in the population, while a cut-off of 79 allowed us to ensure that participants were young enough to engage in physical activity [19]. Additional inclusion criteria were having no complications associated with DM2, having good metabolic control (hbA1c < 7), and showing good adherence to DM2 treatment (defined as adherence to prescribed medication for DM2, physical activity >150 min/week, and healthy diet). The exclusion criteria were gestational diabetes or type 1 diabetes and cognitive impairment. All 10 participants agreed to a follow-up interview. Data saturation [20] was reached by the 10th interview, when we detected that no relevant new information was emerging.

Data were collected using semi-structured interviews (Appendix A File S1). The investigating team prepared a protocol of interview questions which was subsequently adapted in a semi-structured interview form. The interviews were conducted by the principal investigator (PI) from July 2020 to January 2021.

Participants were selected by the referring nurse of diabetes at the primary care centre, who called them and minimally explained the study. If they agreed to participate in the study, the PI contacted them by phone explaining the study in more detail. Once their approval was obtained, participants decided where to conduct the interviews, whether at the patient’s home or a reference health centre. Teams Software was suggested, but the participants opted to conduct the interviews face to face rather than online. Participants were informed of the objectives and asked for their consent for recording the interviews, with all signing the informed consent. Figure 1 shows the sample selection process.

All interviews lasted 35 to 75 min and were audio recorded. Interviews were conducted until data saturation was reached. Sociodemographic data were recorded for all participants and their confidentiality was protected by giving them pseudonyms. The voice files and transcriptions were encrypted and stored on a computer protected with an encrypted password. The interviews were conducted and transcribed textually in Catalan or Spanish, which are both official languages in Catalonia, an autonomous community of Spain. The transcribed interviews were subsequently returned to participants for their approval. All participants accepted their transcribed interviews without changes.

The collected data were recorded, transcribed and analysed using thematic analysis [21,22] with the support of the ATLAS.ti version 9^®^. Figure 2 shows the four steps followed for the thematic analysis.

The patterns identified in the obtained data were systematically organised to answer the questions asked.

The analysis was initiated at the end of the first interviews, as a reflexive process was used. The interviews were conducted, codified and categorised by the principal investigator.

All the stages of the study were discussed with the last author to ensure an accurate/reliable interpretation of the data [23].

The qualitative analysis presented is a complex process as the categories were not pre-established, but gradually built from an inductive–deductive game between the data obtained. The duality of this game allows us to gain a new outlook. Our team was able to go beyond mere description to achieve interpretation through coding and categorising data, as the process used for identifying, organising, refining, relating and integrating the categories required a direct dialectic interaction between the investigator and the data collected, a key factor in qualitative analysis [24].

The trustworthiness of the data was ensured by credibility, dependability, conformability and transferability [15].

Credibility was achieved thanks to analyst triangulation with constant revisions of the themes, subthemes and units of analysis and evaluations, ensuring the qualitative validity of the data by authors 1, 2 and 4. Transferability was confirmed by describing a phenomenon in sufficient detail to be transferable to other settings and populations. Dependability was proven as the findings are consistent and reproducible, and was ensured in this study thanks to the review by the third researcher who examined both the processing and production of the research study. Confirmability was achieved through the reflective effort of each researcher to avoid bias. A transparent description of the research steps taken from the start of the research project was recorded. All methods were accomplished following relevant guidelines and regulations.

The research team has experience in qualitative research and resolving disagreements by consensus, and complied with the Consolidated Criteria for Reporting Qualitative Research (COREQ) [25], with constant revisions made on the thematic analysis process.

## 3. Results

Ten patients with DM2 from four primary care centres in central Catalonia (Spain) participated in the study. Table 1 displays the participants’ main sociodemographic characteristics. Ages ranged from 58 to 79 years, with 60% of participants having had DM2 for more than 10 years; most participants also had a past history of pathology other than DM2.

In our inductive analysis, five categories were identified and grouped into two themes: (1) Integrate PA into lifestyle and (2) Find support for the change of PA, shown in Figure 3.

Table 2 shows an example of the final themes, the codes from which they were built, and an example of a meaning unit from each code.

### 3.1. Integrating PA as a Lifestyle

#### 3.1.1. Meaning of DM2 and PA

The participants’ level of knowledge regarding DM2 varied but all had to learn to live with their disease and acquire new knowledge:

No, no, they told me in a blood test that I was diabetic, and I tell you, I asked the doctor what’s that because I didn’t even know what diabetes was. And I was 57 years old. 5:43 ¶19 (Patient 7)

According to their experience, each patient has a different attitude regarding the causes of DM2, but only one patient thought that PA had caused diabetes. The other participants believe that once you have DM2, PA is important and improves the disease’s prognosis, but they do not think that it caused it:

Well… all sorts of excesses, especially eating and drinking. And not doing any exercise ’cos on top of that I’ve been retired for three years and when I was still active… I was a beast… as I worked for myself, I didn’t have any blood test, that is, I had the first blood test about three years ago when I retired. That’s when I started to roll with all that I’m telling you. 8:4 ¶ 17 (P6)

The other participants believe that their diet, heredity factors and the medication they take mainly caused DM2:

My father, my mother, my grandmother… almost all the family… the maternal one quite a lot and the paternal one, my father… and I think all this influenced it too… They detected it in me when I was 45, 46… now, I’ll be 60… see, it’s been years. 10:4 ¶ 15 (P4)

Other patients do not know how they developed DM2:

I’ve got no idea. Because there isn’t any in my family. I mean I’m the only one in my family. Neither my grandparents had it, nor my parents… (thoughtful) I don’t know… Maybe it’s possible that doing so many diets, now this, now that, and… I think eh? Because I don’t know, it’s something that’s been very hard to swallow hey, diabetes (laughs). 2:4 ¶ 18 (P5)

The meaning the patients give PA is very important and they all agree that it is a key element in their fight against the disease:

I see it very clearly, it’s the diet, and of course the medication too, the medication too of course but the diet and walking are extremely important for a diabetic, super important, very, very important. 5:1 ¶ 7 (P7)

#### 3.1.2. Adaptation to Change and Day-To-Day Strategies

The change of habit of PA is closely linked to the participants seeing that they had a better control over their DM2:

It’s something I did, isn’t it? Everybody tells you, don’t they? That you have to do exercise, you have to walk… but it’s a very personal decision, eh? And though they keep telling you… you’ve got to reach the conclusion that it’s you who’s gotta do it, not for the others, that’s super clear… so it’s you who’s gotta say, it’s a health problem so… if something happens to you you’ll go, the others will stay, you have to look after yourself, love yourself, or if you make up it’s for you, if you go to the hairdresser’s it’s for you, if you have your hair removed it’s for you… I mean not for other people, but for yourself. That’s what I find most important. 10:14 ¶ 71 (P4)

Patients do not perceive that they made changes to start or increase the PA that they currently perform; therefore, the fact that they did not have to alter activities that they were already performing or greatly modify their lifestyles allowed them to adhere to the activity:

Changes… (thoughtful). No change at all, not a single one. Because of course, when I worked I didn’t walk ‘cos I had no time but I retired at 60 and now I’m 70, that means I’ve been doing this for 10 years and, I never gave it up. I didn’t give it up because I feel good, I like it. 5:13 ¶ 43 (P7)

Others who are working thought that work provided them with PA and they did not modify the activities they performed:

I didn’t do it, I haven’t had to make any changes… (thoughtful). No, not much no. One year ago or… but of course then I also moved because I was delivering diaries and went round in a van all day, got out of it, into it… I tell you I moved too. 4:13 ¶ 68–83 (P9)

The majority of patients are retired and many started PA when they stopped work. As they had more time, they could engage in PA whenever they wanted:

But since I retired, I realised that walks were important and for me it’s a habit. I don’t miss a single day and my glucose is fine. 5:2 ¶ 11 (P7)

For the patients who are well adhered to the treatment, the performance of PA is a very important part of their lifestyle:

For me a lot, I feel better when I go out walking than when I stay at home. Because of course me at home all day, ehh… you get used to being at home and you don’t go out. And I prefer to go out than get used to stay sitting down. 4:14 ¶ 84–87 (P9)

The patients mentioned few factors hindering their performance of PA, but it should be borne in mind that they all currently have a good adherence to PA. One of the main obstacles to engaging in PA is the time of day, since this may make it more difficult for them to go out, either because of the heat/cold or darkness:

For me, it’s really hard to go walking in the afternoon. In winter because it’s dark and in summer because it’s very hot and you’re tired of the day and I went some time but not much. 2:2 ¶ 14 (P5)

Others who feel pain find it more difficult to go out:

The older I get, the more my legs hurt. I take the tablets that’s called lyrica and that’s why, so I don’t feel so tired and pain in my bones. If I do too much, even my thighs get stuck, see! 3:34 ¶ 143–146 (P1)

#### 3.1.3. Physical Activity

All the patients are aware of the psychological and physiological benefits of PA:

Well-being. That’s it, sometimes you’re in a bad mood for whatever or sometimes without any reason, and when you go out walking and when you get home, it’s gone. It’s like when you go walking you free yourself, isn’t it? From all the problems, the headaches, you know? And sometimes if you have a problem and you keep thinking about it, when you walk and when you get back it’s like the solution or I think there’s no need to worry… but yes, it helps to free yourself a little. 2:16 ¶ 48 (P5)

The patients are also aware of the benefits of PA regarding their glucose level:

It helps you a lot to control the glucose, a lot. Very important for me, hey? My experience’s proved it. Because of course, I haven’t been diabetic all my life. 5:4 ¶ 15 (P7)

They even feel bad if they do not engage in PA every day:

I feel good. I feel good, because when I don’t do like yesterday… I had a rough day (laughs), yes! That’s the way it is, I feel good. I feel very good. 9:9 ¶ 63(P3)

Most patients walk around town or in the mountains:

Well, walk more. But if I walk yes [] in the morning and in the afternoon. 1:8 ¶ 55–61 (P2)

Some patients also participate in aquagym, a water sport not exclusive to diabetic patients but highly recommended for its benefits:

Two days of aquagym and two or three of walking for about one hour. 3:20 ¶ 85–87 (P1)

Some patients combine two types of PA and even engage in PA twice daily:

Well, in the morning I go for a walk… now I go at about 10, 9.45 till 11 [] and in the afternoon, about 20–25 min doing gym. 4:6 ¶ 34 (P9)

The intensity of PA varies for each patient. They find a pace that allows them to feel fulfilled, each with their own particularities:

Due to my bronchitis no mmm. I can only walk. But I do it regularly, it’s not that I like it… no… I keep a regular pace, regular, you know? 5:11 ¶ 39 (P7)

Sedentarism is very high among diabetic people, but most patients have no or very limited knowledge about this behaviour:

How would you describe sedentary behaviour? (pulls a face as if not understanding what I am saying). Do you know what sedentary behaviour is?—Sanitary?—No… 4:24 ¶ 141–144 (P9)

Look, sedentarism for me is the opposite of what we call willpower and constancy. So, you do what’s easier, from the bed to the armchair, [] you get up, you go to the toilet, you go back to the armchair, and after lunch a good nap. 8:21 ¶ 69 (P6)

Sedentarism is difficult to avoid and the patients do not really know how it could be avoided by diabetic patients:

I don’t know. I’ve often thought of how to motivate people and no, I can’t tell you. I think that… I don’t know, the people… I suppose that for the people who’ve been sedentary all their lives it’s much more difficult and for these people maybe it would be easier to start going out in a group. [] And sometimes you tell them… oh I’ve got a lot of work, oh I don’t want to go with other people… you say we’ll see and you don’t want to go with other people, alone… it’s hard… it’s hard… it’s hard and I don’t know how to do it. 2:24 ¶ 74–76 (P5)

Some think that if other diabetic patients knew the benefits of PA, they would engage in it:

But if they knew… the benefit they’d get… well! They’d think about it, wouldn’t they? A lot, a lot of benefit for the diabetic sure… another could ride a bike. [] For me it’s very good and thanks, I tell you, thanks to my nurse I got used to it and I started going for walks and as I saw that the results were good for me, then I said this works you know? And that cheered me up, yes, yes. 5:22 ¶ 66 (P7)

Others consider that avoiding sedentarism is not so important if the person feels good. They consider that good habits are important once they have been diagnosed, but do not see the importance of preventing it in healthy subjects:

Oh no… if you feel good, why should you avoid it? That it’s not good that people who feel good should do something… but as they feel good, they don’t find it necessary. 3:38 ¶ 162–164 (P1)

### 3.2. Finding Support for a Change of PA

#### 3.2.1. Company Received

Having company when engaging in PA is very important for many patients. Most prefer to go out in very reduced groups and with people in their surroundings, rather than with organised walks involving a greater number of people:

NO! I mean with few people, with a group of very old people as I did sometimes at the clinic they go at their pace, you go at yours… or when you go to the gym you have your pace and they have another… I like to keep my pace and that’s it, but if you find a person or two it’s also very pleasant to go walking and also also. 10:12 ¶ 63 (P4)

However, the majority have to go alone and some even prefer it:

No, some people say that they force you. For me it’s been more… more my own idea and no, I never liked it a lot… (referring to going out with company). 8:13 ¶ 40–42 (P6)

Only one patient thinks that she would not go out if she had to do so alone:

No, if I had to go alone I don’t like it, I don’t like it at all. I find it boring. If I go with a friend and we talk and you don’t realise that you go… sometimes, I have two sisters and we go together the three of us 3. 9:10 ¶ 65 (P3)


Relatives and friends not only accompany them when engaging in PA, but also encourage them to change their previous lifestyle:

My children encouraged me a lot, a lot. They want me to go. They get angry if I don’t. [] If I don’t go it’s because there’s a problem and they can never tell me off because they know that I like it a lot and I never miss it (referring to aquagym). Nobody can tell me off… No, no I don’t feel lazy… but they tell me off to go for walks. Mom, go for a walk, yes I’m going… 3:44 ¶ 140 (P1)

When asked about the involvement of healthcare professionals, some participants initially claimed that they had not received any help to adhere to PA, but later acknowledged their fundamental role:

Maria Àngels, my nurse, helped me a lot to do physical activity. My doctor too but Maria Àngels is the one who looked after me a lot, [] and when there was a talk at the casino and so on she always called me and asked me if I wanted to come… [] And the nurse sure she encouraged me… thanks to her I go on these walks. She started to tell me that walking would be good for me and I thought, maybe she’s right and yes, yes the nurse helped me a lot, a lot, a lot. 5:18 ¶ 53 (P7)

Some patients also emphasised the importance of their general practitioner:

Always the doctors above all don’t give up the pool and above all walk, what you can. Because they know how I am. What you can. Engracia! Whenever you can. 3:63 ¶ 42 (P1)

#### 3.2.2. Personal Motivational Strategies

Patients received external support and comfort to be able to adhere to PA, but the decision to engage in PA or increase adherence was made of their own accord. They saw that it was beneficial to their health and listened to their body. The majority participate in free PA, rather than go with groups. The family has been especially important for them to increase adherence:

Because I myself, sometimes I’m one of these people who think that you have to listen to your body. And I noticed… that… as if the boiler was as if one of these days… maybe will explode now that I’m here with you but then I was really on the edge of everything. Especially drinking… so I more or less gave it up, I had already retired, that was about three and a half years ago… and I thought see, I removed all the drinks from the fridge (…). I took it all out and started to walk a little more seriously… so that’s the two main things I did (…) Yes, yes, all by myself. I thought well, let’s try this. 8:10 ¶ 31–34 (P6)

The awareness that their glycaemia would increase if they did not engage in PA motivated them to continue:

For two months, I didn’t do anything, nothing at all, because I was very tired, and it went up (SUGAR) and I did not change my diet or… so it’s only been PA. 10:18 ¶ 9 (P4)

Each patient found their own motivations to continue performing PA. They sometimes find it difficult to go out and engage in PA but when they do so, they feel physically tired but psychologically better and think they are helping control their DM2. This motivates them to go out:

Sometimes I don’t feel like it, you know? Sometimes you go out and you say I would stay at home not to sleep because I get up early but you say what an effort, you know? And when you come back you feel so good, you feel like new. Of course, depending on the route you take you’re more or less tired but how can I explain, your mood is much better than when you don’t do it. 2:15 ¶ 46 (P5)

The majority engage in PA because they have made the decision themselves; no one has forced them. This fact has been very important in achieving a good adherence:

Good, good, yes, really. Also, they say that exercise releases endorphins in the brain, don’t they? I read that endorphins are released and this causes well-being and well, well… I tell you and the fact that nobody forces me, I have no obligation… […] I think it’s the best attitude for everything, the things that you can decide to do if you want, and so I think it’s good, very good. 6:18 ¶ 72–74 (P8)

When the patients worked they were used to being active, and when they retired they had to look for other activities:

Yes, yes, that’s it. If I stay here on the sofa I’ll get used to the sofa and just stay here. So, you have to go out, and as I was used to going out all day with the van delivering newspapers, delivering magazines… 4:19 ¶ 124 (P9)

They think that PA is extremely important for controlling the disease:

Well for me… (thoughtful), for me (he repeats it firmly) eh? For me it seems that walking is as good as… as… as food and insulin and the tablet. Walking is very important for me, for my experience. It’s… it’s… well you can’t stop it… I walk, I can’t run because I also have chronic bronchitis and if I ran, I’d choke, you know? But I go at my own pace and I walk at least 1.30 h, I keep going, I keep going without stopping but well… I don’t get tired or anything. But of course… at the talks they did at the centre Ateneu… I saw that… that 90% didn’t want to go out for walks, for their age or… or they had other things to do. But as I don’t have anything… it’s OK, it’s perfect for me, walking for a diabetic is… is very… it’s ideal, walking, very good. 5:3 ¶ 13 (P7)

If the patients have friends/family who enjoy PA, it is easier for them to adhere to it:

With those three, my brother, Lluís and Valentí… and these especially these three move a lot… 8:16 ¶ 48 (P6)

Once the patients have acquired the habit of engaging in PA, they feel the need to do so, even if the conditions are not favourable:

Well yes. On the one hand I feel good, and no, I don’t have any… I mean… I don’t have… how can I explain… I don’t feel lazy to say…? These walks don’t… well… and… no, no, and if it rains I take my umbrella and I still go for a walk, eh? And no, no, slowly. I don’t dislike it or anything, I rather like walking. First because, of course, I’m retired and I have all the 24 h in the day, first because the clock goes very fast, it goes very fast. And second I see that for my disease of the glucose it’s perfect. 5:12 ¶ 41(P7)

The patients miss going for walks if they cannot go; it helps them physically and mentally to be able to go out and perform PA:

No, me when I don’t go I miss it. Normally I say, today I’ve just gotta go out because if not it’s as if something’s missing. 2:10 ¶ 34 (P5)

Most of the patients are retired and the performance of PA helps lessen their boredom:

Yes. Yes, because also I feel good, I feel better and more accomplished… not so bored… yes. 9:5 ¶ 47 (P3)

Table 3 highlights and summarizes the essence of the research conducted:

## 4. Discussion

The present study explores the experiences of adaptation and the maintenance of PA in patients with DM2 in a community environment. It also focuses on the individual strategies and environment that allow these people to be considered as examples of a successful adherence to PA.

### 4.1. Integrating PA as a Lifestyle

Daily practice is a key factor for optimal adherence to PA. This degree of daily and constant commitment emphasises the active role of patients when following a treatment prescribed or recommended by a health professional [26].

Depending on their trust in healthcare, their susceptibility and perceived severity is reflected in a desire to live with DM2 better and more healthily, and to avoid progression and further complications due to the disease [27]. This is reflected in the results where patients see improvements in disease control and therefore a better control of complications.

Several published studies focus on newly diagnosed diabetic patients rather than patients with long-term diabetes, and conclude that providing information on diet and PA encourages patients to use PA as a strategy to manage their disease, with most patients considering it useful to make multiple lifestyle changes [28,29].

An insight into the meaning patients attach to DM2 and PA and identification of the causal conditions of DM2 will allow us to know their position and understand their experiences.

Education for self-care should also include information regarding the causes, complications and prognosis of diabetes, and should be adapted to patients’ individual cultural perceptions and needs. The study by Bukhsh et al. found that participants generally had an inadequate knowledge of diabetes, the associated complications and the importance of a healthy diet and regular exercise [30]. The participants in the present study do consider that PA and diet are very important once the disease has been diagnosed, but not for healthy individuals without DM2. Patients’ insufficient knowledge was also considered a barrier, though more seldom [31].

Retirement has shown to be a positive recurrent factor to start and maintain PA. Having more time to engage in PA without having to change their schedule allowed good adherence. A bibliography on retirement and DM2 is scarce. One study mentions the beneficial effects of PA and lifestyle changes on controlling the disease [31]; another mentions lack of time as a common obstacle for performing PA [6]. Others, in line with our study, point out that patients who work do not do perform PA as they are too tired and consider that their daily activities are a fair substitute for PA [30]. The fear of complications caused by a poor control of the disease motivates participants to adhere to the practices of self-care related to diabetes [9,32]. 

The participants in our study do not present complications associated with the performance of PA and consequently found little resistance and/or few problems when doing so. Other studies that have included patients with comorbidities have shown that fatigue, body pain and knee and hip lesions limited their ability to engage in regular exercise [32]. Only one patient in our study referred to occasional pain when engaging in PA, but this did not prevent her from continuing. This may be related to the fact that PA also contributes to a good control of other pathologies associated with a healthy lifestyle, as most participants also present other associated pathologies.

The participants consider that PA is extremely important to control DM2, practise it regularly and experience its benefits, unlike other studies where patients considered medication as significantly more important than diet or physical exercise in controlling diabetes [9]. In the present study, most participants take walks, in agreement with the finding of other studies [32]. Walking is an accessible, cost-free activity that participants can perform whenever convenient. A further benefit of walking is that it is a no-impact activity that does not put undue stress on the joints and the pace can be adapted to each individual. The participants of our study have limited knowledge about sedentary behaviour. This point is of utmost importance as although replacing sedentary behaviour with moderate-to-vigorous activity provides greater health improvements, light activity is also beneficial to metabolic health [33]. Thus, the performance of light exercise in poorly adherent patients could be a temporary solution as it is also beneficial.

### 4.2. Finding Support for PA Change

In line with our findings, recommendations from healthcare providers and support from their family are the key factors that encourage study participants to adhere to diabetes-related self-care practices that include PA [6,34].

Although most physicians and nurses concur in regards providing information, motivation and support to patients to adopt lifestyle changes, as it is part of their duty, just over half consider they have the required skills to provide lifestyle advice. Interestingly, the nurses who had less professional experience reported having more skills than those with more experience. This may be explained by changes in current teaching, which places an increasing emphasis on counselling skills [31]. Other studies have found that regular testing that provides visible feedback and generates trust in the specialised treatment process of diabetic patients contributes to adherence [12], although the patients included in our study were already well adhered to PA, so this would not seem necessary, which stresses the importance of acquiring this habit. The conclusions drawn in a study that looked into family support were very similar to ours. The authors observed that participants mostly received support from their spouse and children, who encouraged them to fight for their health and gave them support by sharing physical activity. Thus, all interventions regarding self-management practices improvement should involve the key members of the family of DM2 patients [9].

A key point in understanding good adherence has been self-care, with the autonomous motivation that patients have developed making it easier for them to make the change permanently; this may be the most important factor in following the recommendations on PA [34], as perceived autonomy support was associated with PA through autonomous motivation. This is in line with the theory of self-determination. Participants seem to gain sufficient energy to maintain an active physical lifestyle once they accept the importance of adequate self-care. A Finnish study focusing on the determinants of performing and maintaining physical activity found close associations between the experience of receiving autonomous support and establishing autonomous motivation towards physical activity [35].

Individuals are very often motivated to adopt and maintain healthy lifestyles by various social factors and supports including policy, system and/or environmental changes [36]. Other studies describe the lack of motivation as an obstacle. Decisions regarding physical exercise, such as a lack of motivation and willpower, as well as not having acquired the habit of engaging in PA, were the most frequently commented barriers to maintaining regular physical activity [37]. Most authors agree that lack of willpower and motivation to adopt lifetime habits is an important barrier for the treatment of the conditions related to lifestyle [31]. It should also be pointed out that studies including interventions to increase PA have determined that the participants were worried about continuing physical exercise alone after the interventions, as they did not find supervised long-term programmes in their communities [38] his was not the case with our sample, as the patients engaged in PA individually or in reduced groups and adhered well to these practices. This highlights the importance of interventions aimed at encouraging autonomous motivation and correct adherence to PA once they are no longer supervised by healthcare professionals.

## 5. Conclusions

A good adherence to PA was achieved thanks to a motivation maintained in time related to the psychological and physiological benefits it brings. External support and care (from families, friends and healthcare professionals) contribute to the adherence to PA and having time to engage in such without making great changes to their day-to-day life once patients have retired. However, the most important recurrent concept lay in the fact that they did not feel forced to make any lifestyle change and their decision was autonomous. This is of utmost importance in achieving a good adherence to PA and is closely connected to autonomous motivation.

Patients’ knowledge of their disease, prevention, treatment and complications is generally superficial and they consider PA as a fundamental pillar once they have developed DM2, but do not consider it important to prevent the disease.

## 6. Limitations

The present study was performed with patients attending primary care centres in a rural region of central Catalonia with specific sociodemographic characteristics. Reproducibility is therefore limited to similar contexts. As our sample is limited, it is not representative of all individuals with DM2 who have adhered to PA. A larger sample of participants from different family, social and cultural backgrounds would strengthen this study, as well as the inclusion of patients who had to make important lifestyle changes. A further limitation is that patients considered retirement as a very important factor to adhere to PA. Although this was not initially foreseen in our study, it would be of great interest to study well-adhered patients who are not retired in order to address further different contexts. It should also be emphasised that this was a qualitative study. No tool was used to evaluate the regularity and intensity of physical activity; the questions asked in the interviews required only subjective declarations. Finally, the present study only focuses on the perspective of individuals with DM2 experiences; they cannot be quantified descriptively and a more complete picture would emerge if the perspective of primary care nurses who follow up these patients had been included.

## 7. Implications for Practice 

This study has enabled the detection of the principal factors related to a correct PA adherence and maintenance by patients with DM2 and how their motivation is maintained in time. This can provide healthcare professionals with the necessary tools to increase long-term adherence to PA. Adherence to PA could be increased by reconsidering the interventions currently applied. Sessions could be conducted to increase patients’ intrinsic motivation, with expert patients (diabetic patients who are well adhered to PA) and professional nurses with specific training to create bonds of affinity between patients. The creation of a virtual platform with well-adhered/expert patients and those who are not would allow them to communicate. A healthcare professional would also have access in order to resolve any doubt that might arise.

## Figures and Tables

**Figure 1 ijerph-19-14041-f001:**
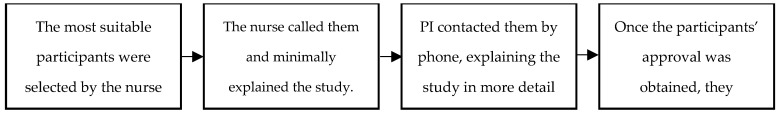
Sample selection process.

**Figure 2 ijerph-19-14041-f002:**
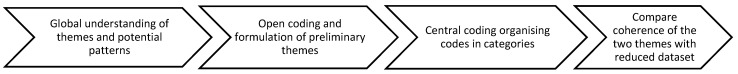
Thematic analysis process.

**Figure 3 ijerph-19-14041-f003:**
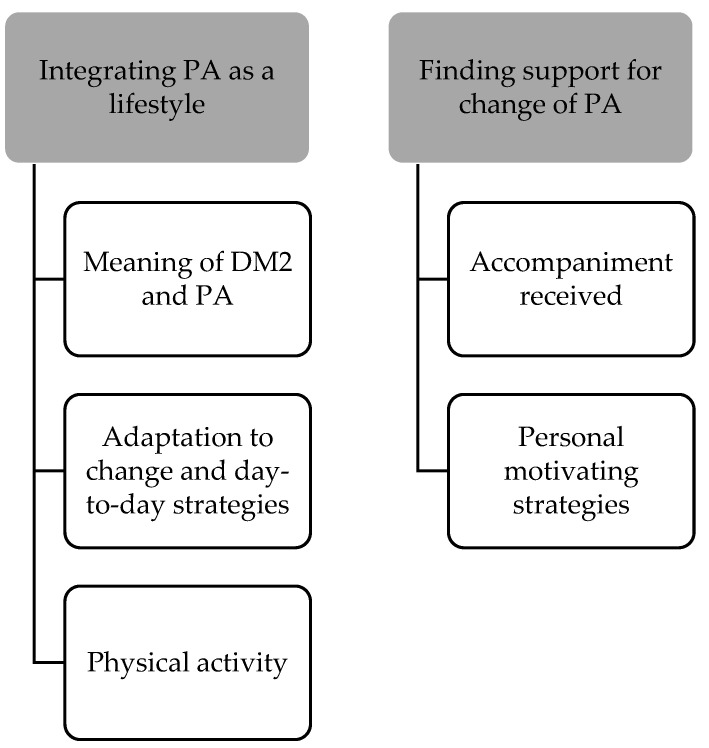
Inductive analysis themes and categories.

**Table 1 ijerph-19-14041-t001:** Sociodemographic characteristics of the participants.

Patient	Age	Gender	Level of Schooling	Employment Status	Years of Evolution of DM2	Relevant Medical History	Marital Status (Widowed, Married)/Living with Partner	HbA1c	Treatment
Patient 1	69	Female	Compulsory education	Retired	18	DM2, obesity, HBP, lumbar arthrosis	Married/living with partner	7	Insulin
Patient 2	76	Male	Compulsory education	Retired	5 (after kidney transplant)	DM2, HBP, mechanical aortic prosthesis, permanent atrial fibrillation, kidney transplant	Married/living with partner	6.2	Oral antidiabetic
Patient 3	70	Female	Compulsory education	Retired	7	DM2, breast cancer, HBP, right knee prosthesis	Married/living with partner	7	Oral antidiabetic
Patient 4	60	Female	Compulsory education	Retired	11	DM2	Married/living with partner	6.9	Oral antidiabetic
Patient 5	62	Female	University degree	Employed	16	DM2, hypothyroidism, Brugada syndrome	Married/living with partner	6.9	Oral antidiabetic and insulin
Patient 6	67	Male	Compulsory education	Retired	8	DM2	Married/living with partner	7	Oral antidiabetic
Patient 7	69	Male	Compulsory education	Retired	10	DM2, COPD, HBP	Widowed/living with daughter	6.7	Oral antidiabetic and insulin
Patient 8	58	Male	Secondary education	Employed	4	DM2, ex-smoker (17 years ago), anxiety, HBP	Unmarried/lives alone	5.5	Oral antidiabetic
Patient 9	79	Male	Compulsory education	Retired	15	DM2, HBP	Married/living with partner	7	Oral antidiabetic
Patient 10	70	Male	University degree	Retired	11	DM2, HBP, generalised seizure in 2014	Unmarried/lives alone	5.7	Oral antidiabetic

**Table 2 ijerph-19-14041-t002:** Inductive analysis themes and codes.

Themes	Codes	Example
Integrating PA as a lifestyle	Causal conditions of DM2	Well man, I think it’s because of the medicine I take for the other thing (referring to other pathologies). 1:3 ¶ 17 (P2)
	DM2 knowledge	But lots of people are not aware, they think that, look, I don’t know, that diabetes is something that’s not for life, till someone tells you that… it’s forever, it’s forever. 10:37 ¶ 130 (P4)
	Personal experience of DM2	I told you that at the beginning I didn’t pay much attention because I didn’t reach dangerous levels, although the doctor told me, you’re diabetic. It can’t be fixed, well… it can’t be fixed… it can’t be cured, hey? So, when you’re a diabetic, it’s for life… mmm… I have the impression that if today I came from I don’t know what country and went to have a blood test and all no one would tell me that I’m a diabetic, of course, now the parameters of the blood test aren’t there. 7:7 ¶ 27–29 (P10)
	Meaning of PA	PA… means… that it’s good for glucose. Second, it’s entertaining, it’s very entertaining, and third I like it, I like it. 5:14 ¶ 45 (P7)
	How the change of PA occurs	Mmm (thoughtful), I don’t know. I think that me, apart from the fact that that you know it in theory, well in practice you see that really it’s really good for you to go walking, at all levels, physical and mental, I say… I mean that… as you see it. As it’s something you don’t see either… for me it’s no… I tell you, it’s no sacrifice to go out. 2:21 ¶ 68 (P5)
	Facilitating factors	I’ve assumed it, I’ve really assumed it that… well… yes… because otherwise… I’m lucky. When you don’t go out, you miss it and you say damn, I haven’t been out for days and I need it, although here in Manresa I do lots of things on foot, but it’s not the same, but it’s not the same than going to do exercise. If you meet someone, I don’t know… I mean I’m not one of those who drives everywhere. 2:14 ¶ 44 (P5)
	Hindering factors	Now go to Manresa and back no way ‘cos I end up going to bed ‘cos everything hurts. Going up hurts and going down too. I can’t do anything about it. That… That’s… 3:59 ¶ 14 (P1)
	Retirement and PA	Well, as I didn’t work, I had all my time free and I could go when I wanted to. 1:23 ¶ 146 (P2)
	New behaviours	Well, you told me to choose, either go to do aquagym at the pool and walk as much as you can because this illness needs a lot of exercise. I’d never done any in my life. 3:56 ¶ 8 (P1)
	Resistance to PA	Precisely, at the time of course I could hardly move. I walked a little with my crutches… then I… I went back to work, I worked for a year and well… with work, my work was physical and about one year ago I caught… I got stuck and from then on I go to the traumatologist and on the waiting list and maybe after the next tests they decide to operate. But they commented that what’s good is walking, that’s why it’s my option and… 6:63 ¶ 106 (P8)
	Benefits of PA	Well, I think that physical activity controls that sugar doesn’t go up, right? [] ‘Cos I think, of course. If I sit on the sofa all day my sugar will go up more, if I move it’ll go up less. 4:5 ¶ 27–30 (P9)
	Sedentary behaviour—knowledge	Mmm… (thoughtful) Well, it’s the fact of finding excuses, any excuse to do nothing. If you don’t want to do anything you find any excuse. 6:24 ¶ 102 (P8)
	Sedentary behaviour—avoid	Well, I don’t know… They should… I don’t know… think, be aware, to say… that’s not working, you know? Change little by little, do… make my own system, you know? Look, instead of sitting down there on the sofa I’m gonna go get the bread every day or the water or whatever you know? And do, do these little things. Because there’s lots of people who, who… I speak to a lot of people who say oh I don’t move from the sofa all day… well that… first that it’s not good at all for their health, you know? And second that I don’t know, hey? You think about things, things, things like why are you here… not me, I don’t have any bad ideas… I entertain myself walking and… now when I get home I’ll go to the supermarket to buy a carton of milk and all that. In the afternoon I go out, I take my daughter’s dog for a walk. 5:24 ¶ 72(P7)
	PA type	And… every day I go and walk, every day, every day. Well maybe not if it’s raining but otherwise, I go every morning after breakfast. And I return… just now at about 11, if not I tell you before I left at about half past 9 and till half past 10 in the Circunvalación Street, a few of us met there and we played cards for a while, but first I went for a walk. 4:1 ¶ 6 (P9)
Finding support for change of PA	PA and company	But I like it more… I’ve even been with a group of friends but I prefer to go by bike at most with a friend… for me to decide what I should do… not that they have to tell me here now… come on come on… now I’m old you can’t control my life… that’s what you sometimes do at the medical centres, you organise walks… I see it and I meet them. Sometimes I meet them… When I went to Cal Aligué I met them crossing the road, hey what are you doing? Maybe I know half a dozen of them who go there and I think… once a week… there’s even a doctor who goes and all in case they get dizzy… they said there was… I don’t need anyone to tell me. Wednesday or Thursday I have to go with them over there. I go there every day! I tell you that every day I walked for two hours… 8:43 ¶ 42 P(6)
	PA and loneliness	That thing about exercise mmm, that’s what I was telling you before that no one forces me, I do things my own way. Sometimes I go and walk round the Parc de l’Agulla three or four times… if you go with someone you have to follow the other one’s pace and so on… I prefer to go alone. 6:58 ¶ 78 (P8)
	Family/friends	My husband, a lot. Because he used to go before me and without him I don’t know if I would have walked or done so much. 2:43 ¶ 56 (P5)
	Influence of professional information	I think, and I don’t want to exaggerate, it’s fundamental… I don’t want to exaggerate. If you find that in this case… in my case I tell you, maybe because we’re work colleagues or… it’s a… I think it’s fundamental that the nurse, above all to bring this unit to the nurse because of course give the different types of people, from very young to very old, and then this… this humanity… Because on top of that it’s one, it’s one nurse who influences your daily life a lot. This human touch to know how to listen to you and speak, I think it’s fundamental for a person in this position. [] She gave me guidelines, she gave me leaflets, she gave me all kind of information ehh… advice… yes, yes, and it’s things that… when we did the last blood test, or whatever it was… this information… 6:27 ¶ 116–120 (P8)
	Motivation to engage in PA	And when you come back you say it’s so damn good, looks like you come back all new. Sure, depending on the route you do you’re more tired but for the mood you feel much better than when you don’t do it. 2:42 ¶ 46 (P5)
	Need to engage in PA	Yes, now that I’m used to it I wanna go out. Even with the heat I go out as well. 1:11 ¶ 77–79 (P2)

**Table 3 ijerph-19-14041-t003:** Themes, categories and essence of the research conducted.

**Integrate PA in lifestyle**	Meaning of DM2 and PA	All the participants had to learn to live with DM2 and acquire new knowledge. They believe that once you have DM2, PA is important and improves the disease’s prognosis, but they do not think that it caused it. The other participants believe that their diet, heredity factors and the medication they take mainly caused DM2. The meaning the patients give PA is very important and they all agree that it is a key element in their fight against the disease.
	Adaptation to change and day-to-day strategies	The change of habit of PA is closely linked to the participants seeing that they had a better control over their DM2. Patients do not perceive that they made changes to start or increase the PA that they currently perform; therefore, the fact that they did not have to alter activities that they were already performing or greatly modify their lifestyles allowed them to adhere to the activity. Participants who worked thought that it provided them with PA. The majority of patients are retired and many started PA when they stopped work (they had more free time). The patients mentioned few factors hindering their performance of PA, but it should be borne in mind that they all currently have a good adherence to PA and that performing PA is a very important part of their lifestyle.
	PA	All the patients are aware of the psychological and physiological benefits of PA and controlling their glucose level. They even feel bad if they do not engage in PA every day. Most patients walk around town or in the mountains. The intensity of PA varies for each patient. They find a pace that allows them to feel fulfilled, each with their own particularities. Sedentarism is difficult to avoid and the patients do not really know how it could be avoided in diabetic patients. Some think that if other diabetic patients knew the benefits of PA, they would engage in it. Others consider that avoiding sedentarism is not so important if the person feels good. They consider that good habits are important once they have been diagnosed, but do not see the importance of preventing it in healthy subjects.
**Find support for the change of PA**	Accompaniment received	Having company when engaging in PA is very important. Most prefer to go out in very reduced groups and with people in their surroundings, rather than with organised walks involving a greater number of people. However, the majority have to go alone and some even prefer it. Relatives and friends not only accompany them when engaging in PA, but also encourage them to change their previous lifestyle. When asked about the involvement of healthcare professionals, some participants initially claimed that they had not received any help to adhere to PA, but later acknowledged their fundamental role.
	Personal motivating strategies	Patients received external support and comfort to be able to adhere to PA, but the decision to engage in PA or increase adherence was made of their own accord. This fact has been very important in achieving good adherence. They saw that it was beneficial to their health and listened to their body. The family has been especially important for them to increase adherence. The awareness that their glycaemia would increase if they did not engage in PA motivated them to continue. They sometimes find it difficult to go out and engage in PA but when they do so, they feel physically tired but psychologically better and think they are helping control their DM2. This motivates them to go out. They think that PA is extremely important for controlling the disease. Once the patients have acquired the habit of engaging in PA, they feel the need to do so, even if the conditions are not favourable. They miss going for walks if they cannot go; it helps them physically and mentally to be able to go out and perform PA. Most of the patients are retired and the performance of PA helps lessen their boredom.

## Data Availability

Not applicable.

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
