# Peer review of "Successful Practices in Performing and Maintaining Physical Activity in Adults with Type 2 Diabetes Mellitus: A Qualitative Study"

_ijerph, 2022, doi:10.3390/ijerph192114041_

Round 1

Reviewer 1 Report

I found an error in the second sentence which stated that 463,000 people have diabetes worldwide in 2019 (Saeedi et al, 2019). The true figure is that 463 million people have diabetes in 2019! Please rectify.

The discussion part is sometimes repetitive and repeats what was mentioned in the results. They can be edited to make them more focus on what authors are trying to convey.

Reviewer 2 Report

Dear Authors the paper needs improvement, especially in the introduction, methodology and presentation of results. the discussion also needs clarification.

1.Introduction

Please further specify in your introduction the recommended amount and type of physical activity for patients with DM2M. Please further specify in your introduction the recommended amount and type of physical activity for patients with DM2M

2. Materials and Methods

as the authors suggest, the study was part of a large project.why, therefore, only a group of 10 participants was selected. What was the sampling procedure and were specific criteria selected? Were there any and if so, what were the exclusion criteria? How many people in total were surveyed and from which sample were these 10 chosen?

how were research participants recruited and were they introduced to the research procedure?

what steps were taken to reduce measurement errors?

Please illustrate in detail, using graphics, the sample selection and the course of the experiment and include it in the procedure subsection.

Did the inclusion criteria used not have an impact on the small size of the research sample?

line 115-please include interview questions protocol in supplementary materials

line 157-159 how was the reproducibility of the procedure and data acquisition determined? were repeat studies carried out or on other groups for verification?

3. Results 

the presentation of the results needs to be improved, they may not be clear to a person not familiar with the subject, it may be worth adding and introducing some tables, diagrams or graphs to highlight the essence of the research carried out

4. Discussion 

line 474-what criteria were used to determine the regularity of physical activity? it should be noted that the answers to the questions given in the interviews are only subjective declarations (it is worth adding this in the limitations paragraph)

Was physical activity assessed in the study using specific tools (smartwatch, miniband, pedometer etc.)?

line 477- was the volume and intensity of physical activity determined according to which criteria?

Was the practised declared physical activity of the study group compared to the recommendations of the diabetes association?

In the discussion, please add a paragraph on the results of studies confirming the fact that factors may affect the lack of motivation to undertake physical activity in people with DM2
